# Evolution of cytokine production capacity in ancient and modern European populations

Jorge Domínguez-Andrés[1,2†], Yunus Kuijpers[3,4†], Olivier B Bakker[5], Martin Jaeger[1,2], Cheng-Jian Xu[1,3,4], Jos WM Van der Meer[1], Mattias Jakobsson[6,7], Jaume Bertranpetit[8], Leo AB Joosten[1,2], Yang Li[1,2,3,4‡*], Mihai G Netea[1,2,9‡*]

[1]Department of Internal Medicine and Radboud Center for Infectious diseases (RCI), Radboud University Nijmegen Medical Centre, Nijmegen, Netherlands; [2]Radboud Institute for Molecular Life Sciences (RIMLS), RadboudUniversity Medical Center, Nijmegen, Netherlands; [3]Department of Computational Biology for Individualised Infection Medicine, Centre for Individualised Infection Medicine (CiiM), a joint venture between Helmholtz-Centre for Infection Research (HZI) and the Hannover Medical School (MHH), Hannover, Germany; [4]TWINCORE, Centre for Experimental and Clinical Infection Research, a joint venture between Helmholtz-Centre for Infection Research (HZI) and the Hannover Medical School (MHH), Hannover, Germany; [5]Department of Genetics, University Medical Centre Groningen, Nijmegen, Netherlands; [6]Human Evolution, Department of Organismal Biology, Uppsala University, Uppsala, Sweden; [7]Centre for Anthropological Research, Department of Anthropology and Development Studies, University of Johannesburg, Auckland Park, South Africa; [8]Institut de Biologia Evolutiva (UPF-CSIC), Universitat Pompeu Fabra, Barcelona, Spain; [9]Department for Genomics & Immunoregulation, Life and Medical Sciences Institute (LIMES), University of Bonn, Bonn, Germany

**\*For correspondence:**
yang.li@helmholtz-hzi.de (YL);
Mihai.Netea@radboudumc.nl
(MGN)

[†]These authors contributed
equally to this work
[‡]These authors also contributed
equally to this work

**Competing interest:** See
page 10

**Reviewing editor:** Jameel Iqbal,
Icahn School of Medicine at
Mount Sinai, United States

**Abstract** As our ancestors migrated throughout different continents, natural selection increased the presence of alleles advantageous in the new environments. Heritable variations that alter the susceptibility to diseases vary with the historical period, the virulence of the infections, and their geographical spread. In this study we built polygenic scores for heritable traits that influence the genetic adaptation in the production of cytokines and immune-mediated disorders, including infectious, inflammatory, and autoimmune diseases, and applied them to the genomes of several ancient European populations. We observed that the advent of the Neolithic was a turning point for immune-mediated traits in Europeans, favoring those alleles linked with the development of tolerance against intracellular pathogens and promoting inflammatory responses against extracellular microbes. These evolutionary patterns are also associated with an increased presence of traits related to inflammatory and auto-immune diseases.

## Introduction

Human history has been shaped by infectious diseases. Human genes, especially host defense genes, have been constantly influenced by the pathogens encountered (*Fumagalli and Sironi, 2014*; *Karlsson et al., 2014*; *Quintana-Murci and Clark, 2013*). Pathogens drive the selection of genetic variants affecting resistance or tolerance to the infection, and heritable variations that increase

survival to diseases with high morbidity and mortality will be naturally selected in people before reproductive age (*Karlsson et al., 2014*). These selection signatures vary with historical period, virulence of the pathogen, and the geographical spread.

Here we investigated the historical evolutionary patterns leading to genetic adaptation in cytokine production and immune-mediated diseases, including infectious, inflammatory, and autoimmune diseases. Cytokine production capacity is a key component of the host defense mechanisms: it induces inflammation, activates phagocytes to eliminate the pathogens and present antigens, and controls induction of T-helper (Th) adaptive immune responses. We have therefore chosen to investigate the evolutionary trajectories of cytokine production capacity in modern human populations during history. To determine the difference in polygenic regulation of diseases and cytokine production capacity, we used data derived from the 500 Functional Genomics (500FG) cohort of the Human Functional Genomics Project (HFGP; http://www.humanfunctionalgenomics.org). The HFGP is an international collaboration aiming to identify the host and environmental factors responsible for the variability of human immune responses in health and disease (*Netea et al., 2016*). Within the HFGP project, the 500FG study generated a large database of immunological, phenotypic, and multi-omics data from a cohort of 534 individuals of Western-European ancestry, which has been used to integrate the impact of genetic and environmental factors on cytokine production and immune parameters. We subsequently deciphered the factors that influence inter-individual variations in the immune responses against different stimuli (*Bakker et al., 2018*; *Li et al., 2016*; *Schirmer et al., 2016*; *Ter Horst et al., 2016*).

## Results and discussion

Peripheral blood mononuclear cells from these individuals were challenged with bacterial, fungal, viral, and non-microbial stimuli, and six cytokines (tumor necrosis factor (TNF)α, interleukin (IL)-1β, IL-6, IL-17, IL-22, and interferon (IFN)γ) were measured at 24 hr or 7 days after stimulation, generating 105 cytokine-stimulation pairs (*Figure 1—figure supplement 1* and *Supplementary file 1a*). The stimulation time intervals were chosen based on extensive studies that showed that the time points used were best suited for assessing monocyte-derived and lymphocyte-derived cytokines per stimulus. Not all the stimuli induce the production of all cytokines; so the selection of the cytokine-stimulus pairs was performed for those pairs for which cytokine production was measurable (*Li et al., 2016*; *Ruschen et al., 1992*; *Schirmer et al., 2016*; *Ter Horst et al., 2016*; *van de Veerdonk et al., 2009*). We correlated cytokine production with genetic variant data to obtain cytokine quantitative trait loci (QTLs), which were employed to compute and compare the polygenic risk score (PRS) of the genomes of 827 individuals from different human historical eras (early upper Paleolithic, late upper Paleolithic, Mesolithic, Neolithic, post-Neolithic), which were downloaded from version 37.2 of the compiled dataset containing unimputed published ancient genotypes (https://reich.hms.harvard.edu/downloadable-genotypes-present-day-and-ancient-dna-data-compiled-published-papers) and 250 modern Europeans randomly selected from the European 1000G cohort. The individuals in this cohort present a similar genetic background (Western-European ancestry) and balanced characteristics in terms of age, sex, body mass index (BMI), and habits (*Ter Horst et al., 2016*), allowing us to represent the natural variability in the immune responses and minimizing the impact of the inter-individual variability in our results. In line with this, cytokines were measured in batches in which a certain cytokine was measured the same day for all the samples, in order to decrease the potential influence of technical variations. Intra-individual variation of cytokine production capacity was found to be limited (*Ter Horst et al., 2021*,) while inter-individual variability was largely depedent on genetic variants (*Bakker et al., 2018*; *Li et al., 2016*).

We investigated the PRS changes over time through constructing linear models and correlation analysis. In order to account for the ancient DNA (aDNA) samples being pseudo-haploid, ambiguous single-nucleotide polymorphisms (SNPs) (A/T and C/G) were excluded when computing PRS to prevent errors due to strand flips. PRS was computed using the most significant QTLs that had a p-value lower than our predetermined threshold for each given trait and removing all variants within a 250-kb window around these variants. Additional PRS models were calculated at varying window sizes (250, 500, 1000 kb) in order to show consistency in the direction of the trends (*Figure 1—figure supplement 2*). The dosage of these variants was multiplied by their effect size while the dosage of missing variants in a sample was supplemented with the average dosage. Although replacing

missing values with the population average does not skew the observed trends over time, it does affect the scale of the data. We opted for this approach, however, due to the scarcity of variants shared by all samples; we focused instead on consistent observations after excluding samples with higher missing genotype rates. Finally, we scaled the PRS to a range of −1 and 1 and correlated the scores of the samples with their respective carbon-dated ages. In order to verify the robustness of our results, we repeated the analysis at multiple threshold combinations for variant missingness and QTL thresholds.

We plotted locally estimated scatterplot smoothing (LOESS) regression models to show the variations in PRS across time for each trait and highlight the time periods around which they drastically change. After plotting the changes before and after the Neolithic revolution at various thresholds for both the missing genotype rate in samples as well as QTL inclusion threshold, only traits that showed consistently significant trends in the same direction were chosen. Additional boxplots and t-tests show the variations in PRS between different pairings besides pre- and post-Neolithic samples. A schematic representation of the steps performed is shown in *Figure 1—figure supplement 3*. Lastly, we used a separate test to determine whether traits were under selective pressure between any two broad time periods using the trait-associated Wright's fixation index (Fst). This test shows an increase in the number of traits under selective pressure after the start of the Neolithic and highlights the traits and points in time for which the observed trends in PRS can be partially attributed to selective pressure as opposed to drift (*Supplementary file 1b*).

Applying the methodology described above, several patterns were apparent (*Figure 1*). The first overall observation is that the estimation of cytokine production capacity based on PRS shows significant differences between populations in various historical periods, and the strength of evolutionary pressure on cytokine responses was different before and after the Neolithic revolution. We did not observe significant changes in cytokine production capacity between individuals who lived in different historical periods before the Neolithic, whereas strong pressure is apparent after adoption of agriculture and animal domestication in Europe. This different pattern may have resulted from the more limited number of samples available for the older time periods, resulting in lower statistical power, but the presence of some evolutionary pressure also before the Neolithic argues that this is most likely not the full explanation. The development of agriculture and domestication of animals in the Neolithic increased population densities on the one hand and the contact between humans and domesticated animals as a source of pathogens on the other hand. The number of zoonoses increased dramatically (examples being tuberculosis, brucellosis, Q-fever, and influenza), which strongly increased the selective pressure and caused significant adaptations of immunity at the genetic level (*Flandroy et al., 2018*). Most of the genetic adaptations to pathogens took place in the period since modern humans abandoned their hunting-gathering lifestyle and developed agriculture (*Deschamps et al., 2016*). In this respect, the strongest changes leading to tolerance (decreased cytokine production) were exerted in the cytokine responses to intracellular zoonotic infections (tuberculosis and *Coxiella*). In contrast, responses to the extracellular pathogens *Staphylococcus aureus* and *Candida albicans* indicate increased resistance, with high production of IL-22 and TNFα, respectively. The increased response to the important fungal pathogen *C. albicans* after the Neolithic period is validated also at the transcriptional level. Overall, these patterns are reminiscent of the studies showing that human immune responses need to adapt to a new landscape of infectious agents depending on the geographical location and types of microbe encountered (*Ferwerda et al., 2007*). Such different patterns were most likely encountered also through history.

Importantly, our results also show significant patterns in changes in the production of specific cytokines during history. The resistance against intracellular pathogens increased after Neolithic with higher IFNγ responses (see *Figure 1—figure supplement 1*): indeed, it is known that Th1-IFNγ responses are crucial for the host defense against intracellular pathogens such as mycobacteria or *Coxiella* (*Thakur et al., 2019*). In addition, the resistance to the extracellular pathogens *C. albicans* and *S. aureus* is also increased after this Neolithic era, with TNFα and IFNγ production increasing steadily after. These two cytokines are very well known to be important for anti-*Candida* and anti-*Staphylococcus* host defense (*MRSA Systems Immunobiology Group et al., 2018*; *Domínguez-Andrés et al., 2017*). On the other hand, a different pattern emerges in relation with the IL1β-IL6-IL17 axis: the production of these cytokines is seen decreasing after Neolithic (see *Figure 1a and b*). In this context, the decrease through time of poly I:C induction of cytokines, as a model of viral stimulation, is intriguing but potentially very important: many important viruses such as influenza and

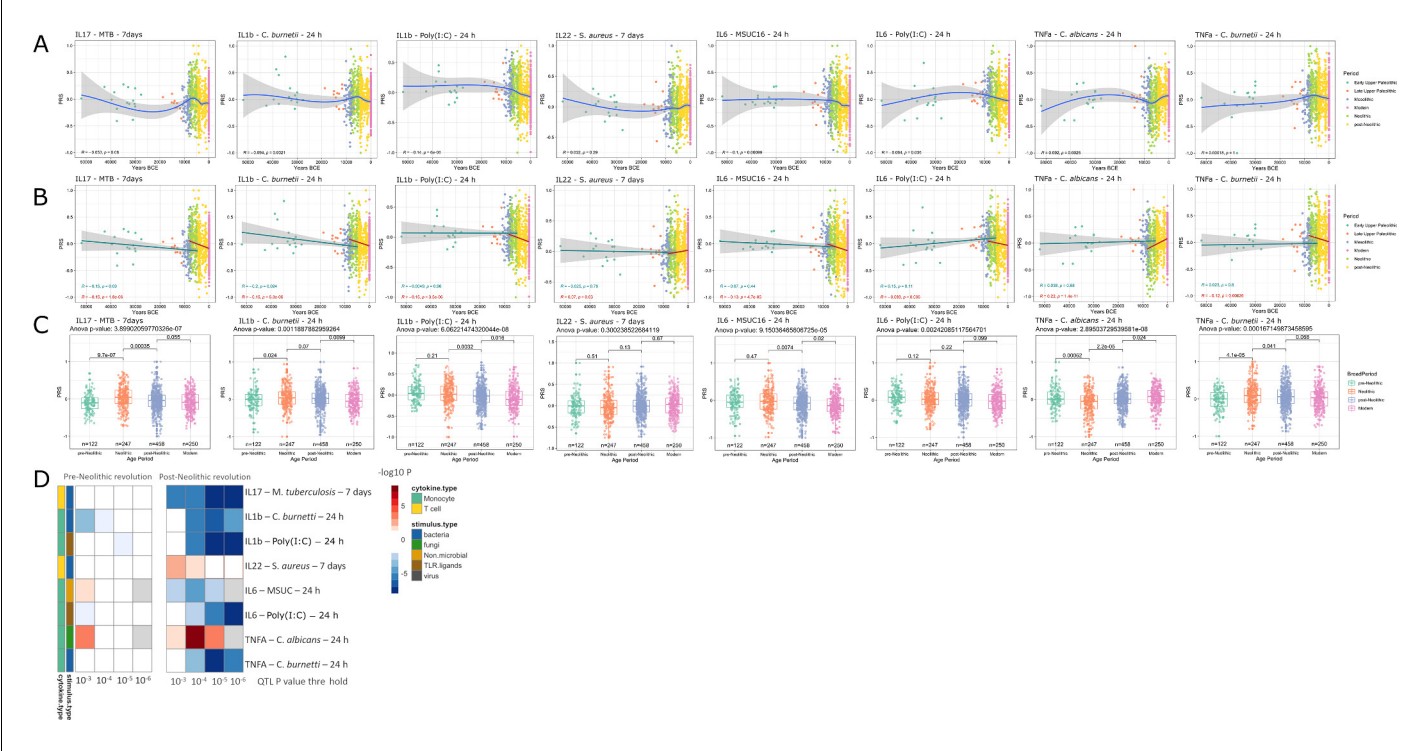

**Figure 1.** Distribution of PRS of cytokine response QTLs across time. (**A–C**) Polygenic risk score (PRS) models based on quantitative trait loci (QTLs) at a p-value threshold of 1e-5. Gray area around regression lines represents the 95% confidence interval. (**A**) LOESS regression models showing the changing differences in PRS across time. (**B**) Dual linear models showing the difference in trends before and after the Neolithic revolution. (**C**) Boxplots showing the difference in mean PRS between adjacent broad time periods using a t-test and the overall difference in means using analysis of variance (ANOVA). (**D**) Heatmap showing the consistency in regression trends before and after the Neolithic revolution using different QTL inclusion thresholds (1e-3 to 1e-6). Intracellular organisms that can cause zoonotic disease: *Mycobacterium* tuberculosis, *Coxiella burnetti*. Extracellular organisms: *Staphylococcus aureus*, *Candida albicans*. MSUC: monosodium urate crystals. PolyIC: polyinosinic:polycytidylic acid.

The online version of this article includes the following figure supplement(s) for figure 1:

**Figure supplement 1.** Correlation between cytokine PRS and time.

**Figure supplement 2.** PRS correlation pre- and post-Neolithic revolution using polygenic scores calculated at various window sizes.

**Figure supplement 3.** aDNA and modern DNA samples of European individuals were used in combination with summary statistics from predominantly.

coronaviruses (severe acute respiratory syndrome (SARS), Middle East respiratory syndrome (MERS), and SARS coronavirus 2 (SARS-CoV-2)) exert life-threatening effects through induction of cytokine-mediated hyperinflammation (also termed 'cytokine storm') (*Tay et al., 2020*); evolutionary processes to curtail these exaggerated responses are thus likely to be protective, and tolerance against viruses becomes a host defense mechanism (*Diard and Hardt, 2017*).

These evolutionary genetic adaptations to pathogens throughout human history greatly influence the way we respond to multiple diseases in modern times as well. To assess these effects, we calculated the PRS associated with the risk of several highly prevalent immune-mediated diseases. The first focus was on common infectious diseases such as malaria, human immunodeficiency virus-acquired immunodeficiency syndrome (HIV-AIDS), tuberculosis, and chronic viral hepatitis; we calculated the changes in susceptibility to these diseases in the last 50,000 years of human history, based on summary statistics from genome-wide association studies (GWAS) databases available from the literature (*Figure 2—figure supplement 1*). Our results show that humans are becoming more resistant to these diseases, with the notable exception of tuberculosis, whose risk score remained stable along the period studied (*Figure 2*). Of note, the QTLs that passed our thresholds were scarce, resulting in a PRS model using a limited number of SNPs, making them sensitive to changes in the p-value cutoff, especially in traits related with infectious diseases. Our results suggest that humans have built up a genetic makeup that made them more resistant to a variety of microbes. The pattern of this adaptation is very interesting as well, with a suggested decrease of susceptibility to malaria

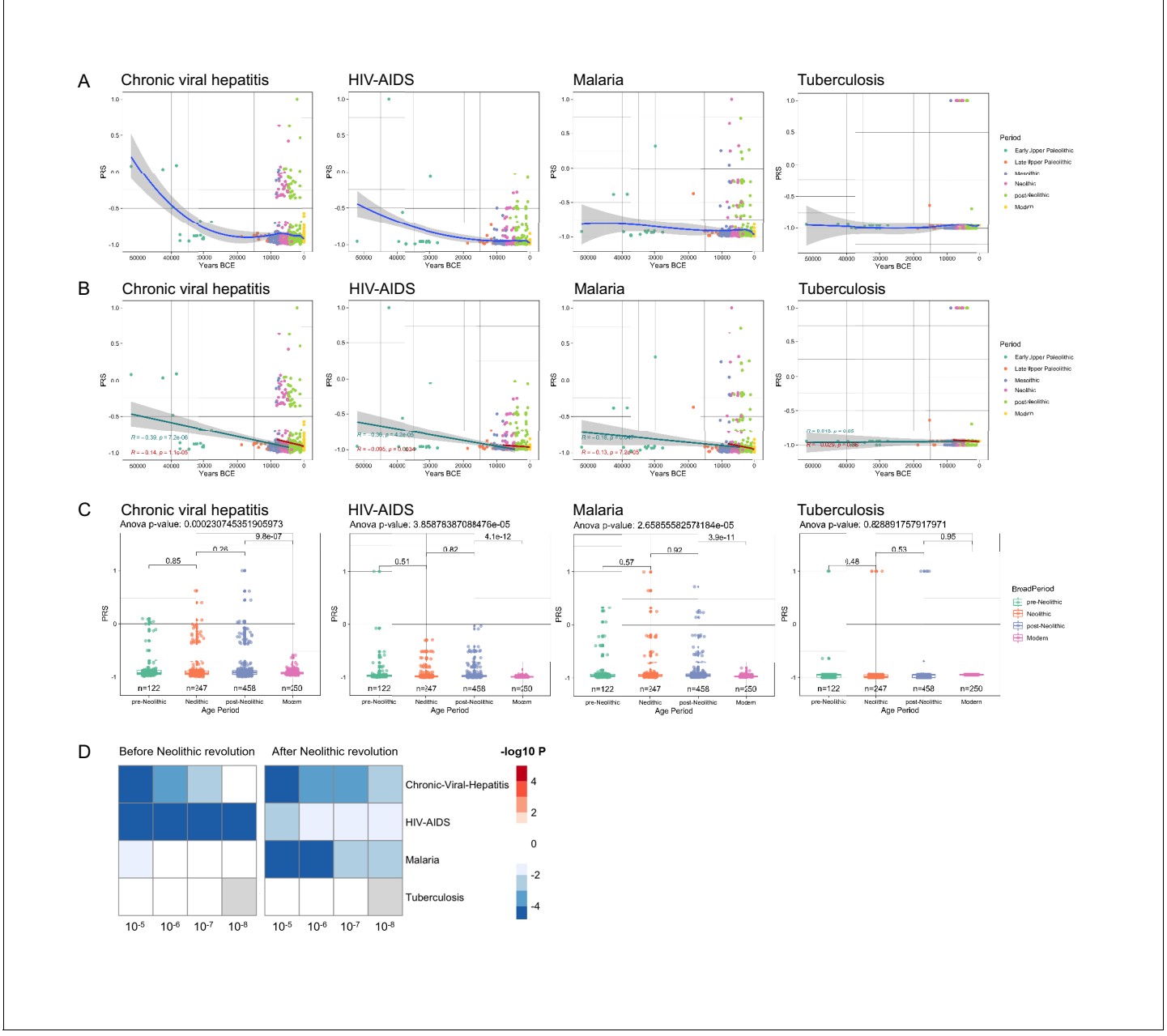

**Figure 2.** Distribution of PRS of infectious diseases across time. (A–C) Polygenic risk score (PRS) models based on a p-value threshold of 1e-5. Gray area around regression lines represents the 95% confidence interval. (A) LOESS regression models showing the changing differences in PRS across time. (B) Dual linear models showing the difference in trends before and after the Neolithic revolution. (C) Boxplots showing the difference in mean PRS between adjacent broad time periods using a t-test and the overall difference in means using analysis of variance (ANOVA). (D) Heatmap showing the consistency in regression trends before and after the Neolithic revolution using different quantitative trait locus (QTL) inclusion thresholds (1e-5 to 1e-8). The online version of this article includes the following figure supplement(s) for figure 2:

**Figure supplement 1.** GWAS summary statistics and cohorts used for PRS calculation.

especially in the last 10,000 years. The reason for this accelerated resistance after Neolithic might be linked to a higher disease prevalence due to increased population density, as otherwise *Plasmodium* parasites are known to have circulated in Africa since at least the Paleogene 30 million years ago (*Poinar, 2005*), and we have likely inherited it from gorillas (*Liu et al., 2010*). Intriguingly, we also observed a strong decrease in susceptibility to HIV: this is a contemporary pathogen, therefore this signal could be due to common genetic and immune pathways with other infections that were

present in human populations. The increased resistance to HIV in Europeans may be derived from selective pressures induced by other pathogens such as *Yersinia pestis* (*Duncan et al., 2005*). Our data suggest, on the other hand, that the source of this increased resistance is even older.

In contrast, the lack of genetic adaptation in susceptibility to tuberculosis is intriguing. This surprising finding may be explained by a concept in which *Mycobacterium tuberculosis* is at the same time a pathogen and a symbiont, in which latent infection enhances the resistance against other pathogens, and this is why our immune system tolerates mycobacterial presence (*Pai et al., 2016*). In this regard, individuals with latent tuberculosis exhibit enhanced macrophage functions that may protect against other pathogens through the induction of trained immunity (*Joosten et al., 2018*). In this context, humanity may not be adapting to tuberculosis because increased resistance against mycobacteria is not evolutionarily advantageous. All in all, these results suggest that the risk of suffering infectious diseases has steadily decreased at least for the last 50,000 years as a result of the selection of genetic variants that confer resistance to infections.

It has been proposed that the increased prevalence of inflammatory and autoimmune diseases is associated with the immune-related alleles that have been positively selected through evolutionary processes to protect against infection; hence, the contrasting differences in the prevalence of auto-immune diseases between populations result from diverse selective pressures (*Ramos et al., 2015*). In line with this, it has been hypothesized that genetic variants associated with protection against infectious agents are behind the increased prevalence of autoimmune diseases in populations with low pathogen exposure, such as Europeans (*Fumagalli et al., 2011*; *Raj et al., 2013*). To study the changing patterns of susceptibility to autoimmune and inflammatory diseases during history, we used publicly available summary statistics from GWAS of digestive tract-related autoimmune and inflammatory diseases and arthritis-related diseases (see *Figure 2—figure supplement 1*) and calculated the PRS for each of the samples under study. Interestingly, we observed a robust increase of the genetic variants related with the development of inflammatory diseases in the digestive tract after the Neolithic revolution (*Figure 3*). PRS scores associated with celiac disease, Crohn's disease, ulcerative colitis, and inflammatory bowel disease were strongly associated with the age of the samples, regardless of the p-value thresholds or the missing genotype rates used for PRS calculation, showing the robustness of these results (see *Figure 1—figure supplement 2*). The fact that especially intestinal inflammatory pathology is increased after a historical event that fundamentally modified human diet is unlikely to be an accident. Our results are in line with earlier research demonstrating that variants in genes important for immune responses and involved in celiac disease pathophysiology (such as IL-12, IL-18RAP, SH2B3) are under strong positive selection (*Zhernakova et al., 2010*). The reasons for the selection pressure on these genes are not completely understood, but an advantage for host defense has been suggested (*Zhernakova et al., 2010*).

In contrast to intestinal inflammation, the PRS of traits linked with juvenile-idiopathic arthritis, rheumatoid arthritis, and multiple sclerosis shows a decrease in genetic susceptibility with the age of the sample after the Neolithic revolution. For pre-Neolithic periods, these patterns had little impact with decreasing PRS for digestive tract diseases and increasing PRS for ankylosing spondylitis and juvenile idiopathic arthritis. A strong decrease in susceptibility to juvenile idiopathic arthritis, rheumatoid arthritis, and multiple sclerosis is seen after the Neolithic period (see *Figure 3*). This is likely linked to the decreased production of the IL-1/IL-6/IL-17 axis described in *Figure 2*, which is particularly important in the pathophysiology of these disorders (*Akioka, 2019*; *Mei et al., 2011*).

The significant changes in cytokine production and disease susceptibility in European populations after the Neolithic can be due to selective processes on the one hand (as described above), but also due to important demographic changes due to migrations of human communities such as the Anatolians (in Neolithic) or the Yamnaya populations from the Pontic steppe (during the Bronze Age) (*Racimo et al., 2020*). In this regard, several loci associated with inflammatory diseases displayed a group of alleles linked with Crohn's disease, celiac disease, and ulcerative colitis in Neolithic Aegeans, the community that spread farming across Europe (*Hofmanová et al., 2016*), with several of these alleles showing signs of positive selection in modern Europeans (*Raj et al., 2013*). In addition, the gene expression PRS of several cytokines based on the cis- and trans- [gene] expression quantitative trait locus (eQTLs) from the eQTLGen Consortium (https://www.eqtlgen.org/) displayed a very strong association with time for TNFα after the Neolithic revolution (*Figure 4*).

Of note, the availability of samples in the pre-Neolithic dataset is limited compared to the post-Neolithic era. To test the robustness of the trends observed, we recalculated the correlation

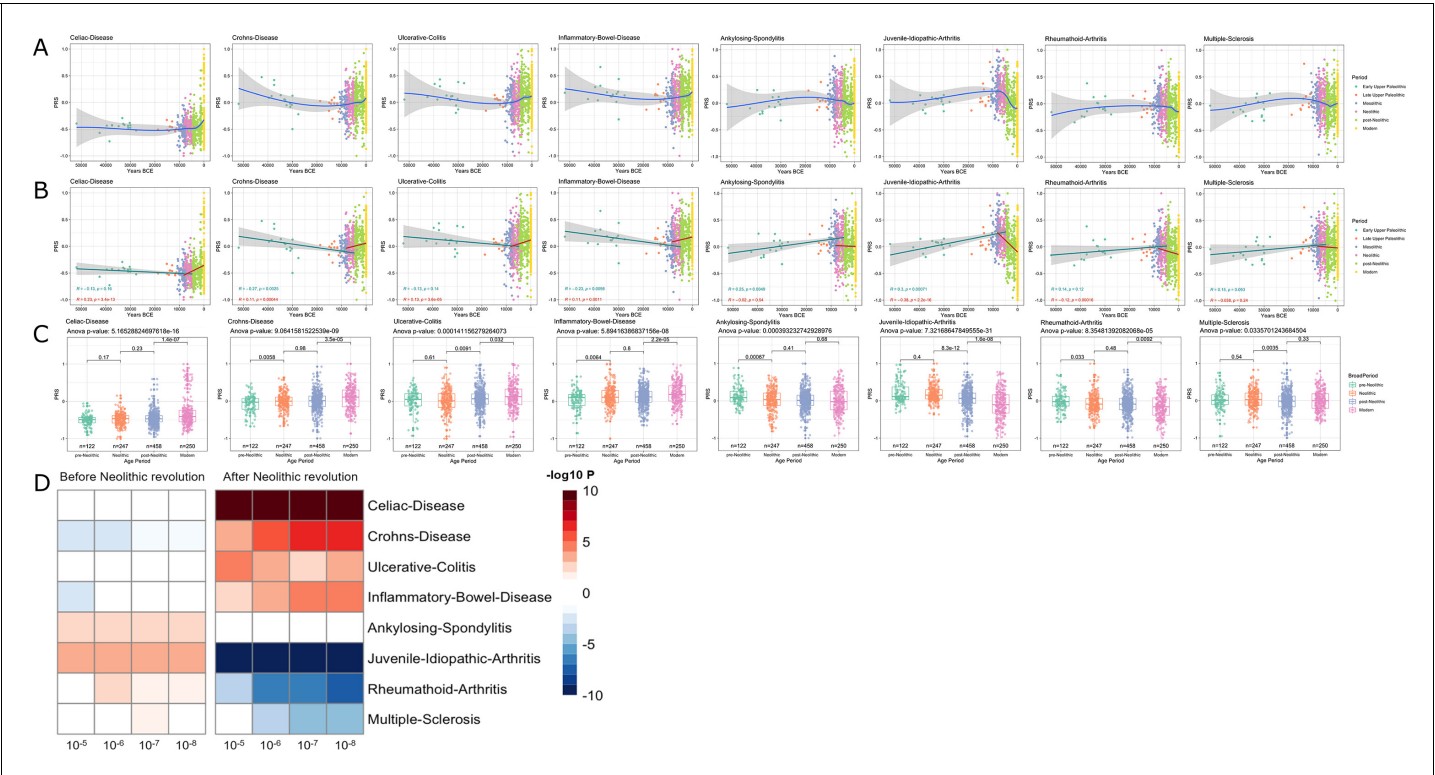

**Figure 3.** Distribution of PRS of inflammatory diseases across time. (A–C) Polygenic risk score (PRS) models based on a p-value threshold of 1e-5. Gray area around regression lines represents the 95% confidence interval. (A) LOESS regression models showing the changing differences in PRS across time. (B) Dual linear models showing the difference in trends before and after the Neolithic revolution. (C) Boxplots showing the difference in mean PRS between adjacent broad time periods using a t-test and the overall difference in means using analysis of variance (ANOVA). (D) Heatmap showing the consistency in regression trends before and after the Neolithic revolution using different quantitative trait locus (QTL) inclusion thresholds (1e-5 to 1e-8).

coefficients and p-values to show the consistency of the results from the down-sampled data of the post-Neolithic samples to match the sample size of the pre-Neolithic samples. This analysis showed that the trajectories observed in our results are consistent regardless of the sample size (*Figure 4— figure supplement 1*).

In this study we focus on the effects of different pathogens on cytokine production and, through that, on the response to different infections. Human evolution has been influenced by multiple factors such as migration, urbanization, climate, and diet, which also influence the responses to pathogens. The advent of the Neolithic lifestyle was a milestone for human societies: after the Neolithic, the density of human populations and the rate of contact with domesticated animals increased significantly, which increased the emergence of pathogens and the ease with which those could spread. Our data show that while various events and conditions throughout human history have influenced our immune responses to pathogens, the advent of the Neolithic lifestyle was an important turning point for our capacity to respond to pathogens.

Collectively, our results show that the advent of the Neolithic era was a turning point for the evolution of immune-mediated traits in European populations, driving the expansion of alleles that favor the development of tolerance against intracellular pathogens and promote inflammatory responses against extracellular microbes. This is associated with a higher presence of genetic traits related with inflammatory and auto-immune diseases of the digestive tract and a lower number of alleles linked with the development of arthritis. It is important to underline that our results are obtained using European aDNA samples and GWAS summary statistics computed using European cohorts. Therefore, our results would be specific to European populations and should be interpreted with caution for populations from other geographical locations. Further research should compare the trends in different populations that have been exposed to different environments across the planet and clarify

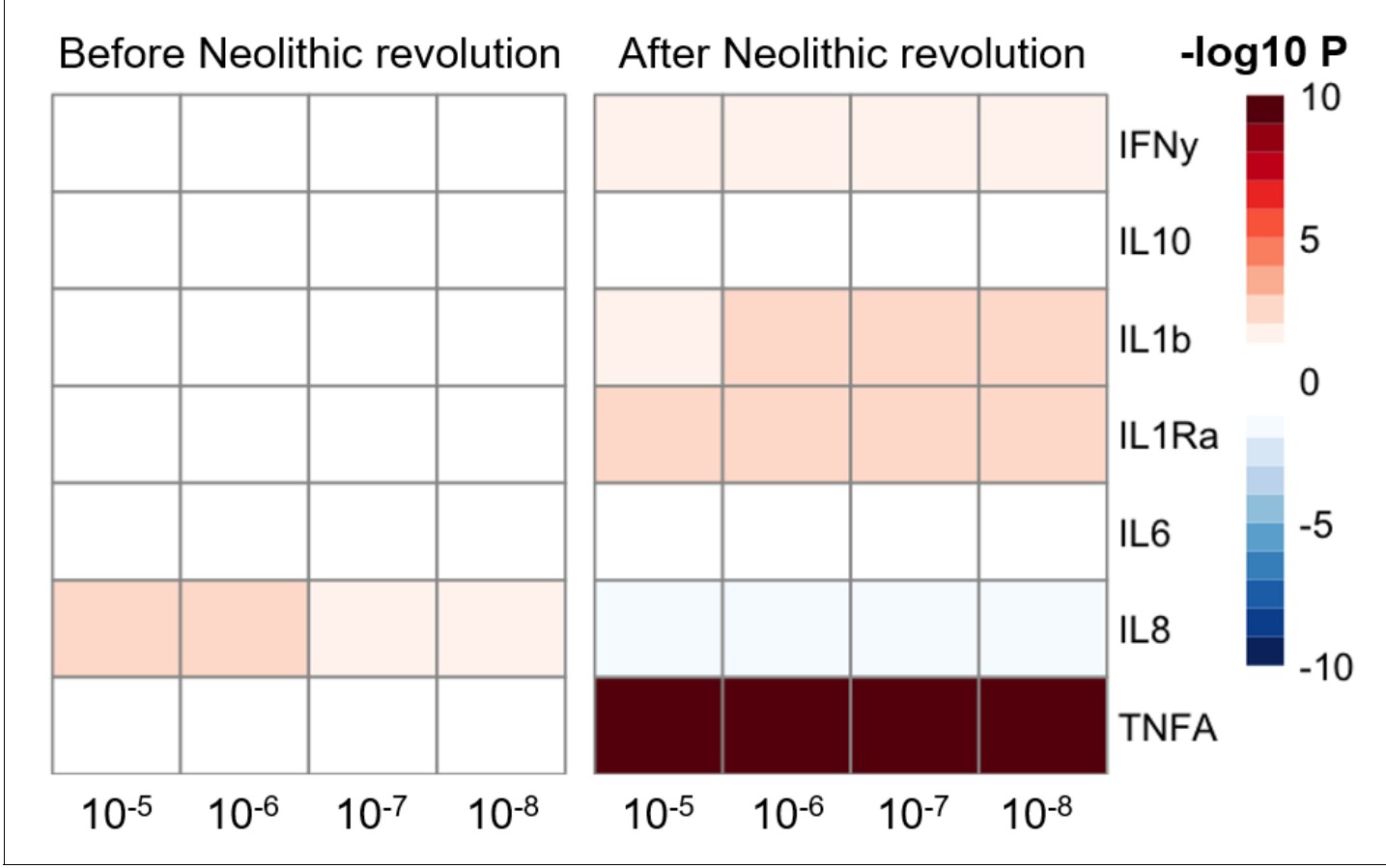

**Figure 4.** Changes in cytokine gene expression PRS across time before and after the Neolithic revolution using different QTL thresholds (1e-5 to 1e-8). Missing genotype rates ranged through 0.96, 0.9, 0.8, and 0.7. Quantitative trait locus (QTL) p-values for variants included in our polygenic risk score (PRS) models ranged through $10^{-3}$, $10^{-4}$, $10^{-5}$, $10^{-6}$, $10^{-7}$, and $10^{-8}$. The color key indicates the range of -log10 p-values of the Pearson correlation between PRS and time. Red and blue indicate positive and negative association, respectively.

The online version of this article includes the following figure supplement(s) for figure 4:

**Figure supplement 1.** Robustness of correlation coefficients post-Neolithic is independent of the sample size.

the influence of ancestry, time, and rural vs urban lifestyle to shed light on the influence of the infectious environment on genetics and human evolution.

## Materials and methods

### Cohort selection

Ancient DNA genotype data were downloaded from version 37.2 of the published aDNA genotype database, compiled by and available on the David Reich Lab website (https://reich.hms.harvard.edu/downloadable-genotypes-present-day-and-ancient-dna-data-compiled-published-papers). The aDNA samples consisted of pseudo-haploid genotype data. This was due to the low genotyping coverage. Samples with variant missingness above 96% were filtered out using Plink (*Purcell et al., 2007*). This was done in order to remove outliers with extremely low coverage. Only samples within Europe were used for this study, and these samples were selected based on their geographic location, that is latitude (within 35 and 70 degrees north) and longitude (within 10 degrees west and 40 degrees east). Samples without a carbon-dated age were also filtered out. We also selected 250 European samples from the 1000 genomes project phase 3. Only variants present in both the ancient samples and the modern samples were retained. This resulted in a dataset of 827 ancient samples and 250 modern samples containing 1,233,013 variants.

## Carbon-dated sample origin and geographical location

Both carbon-dated age of origin and latitudinal and longitudinal data were available for these 827 ancient European samples. Broad time periods were assigned to these samples with the early upper Paleolithic era for all samples originating from before 25,000 years before the common era standardized to 1950 (BCE). The late upper Paleolithic era follows until 11,000 BCE. The Mesolithic era ranges from 11,000 to 5500 BCE. The Neolithic era ranges from 8500 to 3900 BCE, and the post-Neolithic era ranges from 5000 BCE and more recent ages. Using the geographical data in combination with archeological clues and the genetic data, the broad time period of origin was also available for samples that were dated to a point in time with overlapping broad time periods. This allowed the samples to be classified as either early upper Paleolithic, late upper Paleolithic, Mesolithic, Neolithic, or post-Neolithic. The sample age of the 250 modern European samples was set to 0.

## Summary statistics of GWAS and cytokine QTLs

Summary statistics for complex traits were obtained from the UK Biobank (*Bycroft et al., 2018*) and the GWAS catalog (*MacArthur et al., 2017*) last accessed on March 29, 2020. The stimulated cytokine response summary statistics from the 500FG cohort of the HFGP were used (*Li et al., 2016*). Some complex traits had multiple different sets of summary statistics available. In these cases, the data, which were more recent and used bigger cohorts that were either of European or mixed (European and Asian) ancestry, were selected. The variants of these summary statistics were then filtered by only keeping bi-allelic variants. Most aDNA genotypes available are pseudo-haploid as a consequence of their lower sample quality. We excluded ambiguous SNPs (A/T and C/G) in order to prevent errors due to strand flips present in these pseudo-haploid samples.

## Polygenic risk score calculation

Polygenic risk scores were then calculated by first intersecting the filtered variants from the summary statistics with the variants present in the DNA samples. Starting at the most significant variant, all variants within a 250-kb window around that variant were excluded until no variants remained. We then multiplied the dosage of these variants with the effect size and these values were summed. If a variant is missing in a sample, the dosage is substituted with the average genotyped dosage for that variant within the entire dataset. This way the PRS is not skewed in any specific direction. The formula for this is described below with the score $S$ being the weighted sum of a variant's dosage $X_n$ multiplied by its associated weight or $\beta_n$ calculated using $m$ variants.

$$S = \sum_{n=1}^{m} X_n \beta_n$$

## Relation between PRS and carbon-dated sample age

We constructed piecewise linear models for each trait by separating the samples into two groups. These two groups consisted of all samples preceding the Neolithic era and those of the Neolithic era and later, respectively. We correlated PRS with the carbon-dated age of our samples. We then multiplied the -log10 of the Pearson correlation p-values with the sign of the correlation coefficients.

In addition, we plotted LOESS regression models to highlight the change in PRS at each point in time independent of any predefined breakpoint between historical periods. We also performed a group-based comparison using Student's t-test. We compared pre-Neolithic, Neolithic, post-Neolithic, and Mesolithic samples with their respective adjacent historical periods to show the difference between other historical transitions besides the pre- and post-Neolithic.

## Selection test

We tested whether traits were under selection or if observed changes were due to the genetic drift between adjacent time periods. We performed a two-tailed test using the mean $F_{st}$ calculated with trait-specific SNPs between two adjacent periods and a reference distribution of 10,000 random linkage disequilibrium (LD)- and minor allele frequency (MAF)-matched mean $F_{st}$ scores calculated using an equal amount of SNPs (*Schirmer et al., 2016*). Bonferroni correction was performed to account for multiple testing.

## Robustness of results

In order to test the robustness of our results we calculated PRS using multiple different p-value thresholds for QTL inclusion. We used p-value thresholds from $10^{-3}$ to $10^{-8}$ for the complex traits obtained through GWAS catalog and the UK Biobank. The thresholds used for the stimulated cytokine responses ranged from $10^{-3}$ to $10^{-6}$. We also calculated PRS using different variant missingness thresholds. This means we removed samples with a variant missingness rate higher than 96, 90, 80, or 70%. All of the results from the piecewise linear models were then used to create a heatmap depicting the consistency and robustness of our observed correlations.

Additionally, various window sizes were used for clumping the QTLs, and LD-based clumping was also performed excluding variants with an LD greater than 0.2 compared to our lead SNP within a window. In order to see whether our observations were due to sample imbalances between the pre-Neolithic period and the later periods, samples originating from the Neolithic period and later were randomly down-sampled to the same number of samples as the pre-Neolithic samples. Correlation coefficients between PRS and sample age were then recalculated for the Neolithic and younger samples and compared to the coefficients obtained using all Neolithic and younger samples.

## Acknowledgements

MGN was supported by an ERC Advanced Grant (833247) and a Spinoza Grant of the Netherlands Organization for Scientific Research. YL was supported by an ERC Starting Grant (948207) and the Radboud University Medical Centre Hypatia Grant (2018) for Scientific Research. JD-A is supported by The Netherlands Organization for Scientific Research (VENI grant 09150161910024). JB was supported by PID2019-110933GB-I00 (AEI/FEDER, UE) MINECO, Spain.

## Additional information

### Competing interests

Jos WM Van der Meer: Senior editor, *eLife*. The other authors declare that no competing interests exist.

### Funding

| Funder | Grant reference number | Author |
|---|---|---|
| European Commission | 833247 | Mihai G Netea |
| European Commission | 948207 | Yang Li |
| ZonMw | 09150161910024 | Jorge Domínguez-Andrés |
| Netherlands Organisation for Scientific Research | Spinoza | Mihai G Netea |
| Radboud University Medical Center | Hypatia Grant (2018) | Yang Li |
| Netherlands Organisation for Scientific Research | 09150161910024 | Jorge Domínguez-Andrés |
| MINECO | PID2019-110933GB-I00 (AEI/FEDER UE) | Jaume Bertranpetit |

The funders had no role in study design, data collection and interpretation, or the decision to submit the work for publication.

### Author contributions

Jorge Domínguez-Andrés, Conceptualization, Investigation, Writing - original draft; Yunus Kuijpers, Conceptualization, Data curation, Formal analysis, Investigation, Methodology, Writing - original draft; Olivier B Bakker, Resources, Formal analysis, Investigation; Martin Jaeger, Resources, Writing - review and editing; Cheng-Jian Xu, Resources, Methodology, Writing - review and editing; Jos WM Van der Meer, Supervision, Writing - review and editing; Mattias Jakobsson, Resources,

Methodology; Jaume Bertranpetit, Resources, Supervision, Writing - review and editing; Leo AB Joosten, Resources, Supervision; Yang Li, Mihai G Netea, Conceptualization, Resources, Supervision, Funding acquisition, Project administration, Writing - review and editing

### Author ORCIDs
Jorge Domínguez-Andrés (iD) https://orcid.org/0000-0002-9091-1961
Yunus Kuijpers (iD) https://orcid.org/0000-0002-5075-3970
Cheng-Jian Xu (iD) https://orcid.org/0000-0003-1586-4672
Leo AB Joosten (iD) http://orcid.org/0000-0001-6166-9830
Yang Li (iD) https://orcid.org/0000-0003-4022-7341
Mihai G Netea (iD) https://orcid.org/0000-0003-2421-6052

### Decision letter and Author response
Decision letter https://doi.org/10.7554/eLife.64971.sa1
Author response https://doi.org/10.7554/eLife.64971.sa2

## Additional files

### Supplementary files
• Supplementary file 1. Stimulated cytokine expression combinations at different timepoints. (**a**) Overview of the stimulus, cytokine, and time-point combinations. In total, 105 unique stimulated cytokine traits were available using various types of stimuli measuring both the innate and the adaptive immune response. (b) Tests for selection as opposed to the genetic drift were performed between populations from adjacent time periods. A two-tailed test was used to determine whether the mean trait Fst between pre-Neolithic–Neolithic, Neolithic–post-Neolithic, and post-Neolithic–Modern samples was significantly different compared to 10,000 random LD- and MAF-matched mean Fsts calculated using the same amount of single-nucleotide polymorphisms (SNPs).

• Transparent reporting form

### Data availability
All the data employed in this manuscript have been obtained from publicly available databases.

The following previously published datasets were used:

| Author(s) | Year | Dataset title | Dataset URL | Database and Identifier |
|---|---|---|---|---|
| Trynka G, Hunt KA, Bockett NA, Romanos J, Mistry V, Szperl A, Bakker SF, Bardella MT, Bhaw-Rosun L, Castillejo G, Almeida RC, Dias KR, Dubois PC, Duerr RH, Edkins S, Franke L, Fransen K, Gutierrez J, Heap GA, Hrdlickova B, Hunt S, Izzo V, Joosten LA, Langford C, Mazzilli MC, Mein CA, Midah V, Mitrovic M, Mora B | 2011 | Dense genotyping identifies and localizes multiple common and rare variant association signals in celiac disease | https://www.ebi.ac.uk/gwas/publications/22057235 | NHGRI-EBI GWAS Catalog, 22057235 |
| Liu JZ, Huang H, Ng SC, Alberts R, Takahashi A, Ripke S, Lee JC, Jostins L, Shah T, Abedian S, Cheon JH, Cho | 2015 | Association analyses identify 38 susceptibility loci for inflammatory bowel disease and highlight shared genetic risk across populations | https://www.ebi.ac.uk/gwas/publications/26192919 | NHGRI-EBI GWAS Catalog, 26192919 |

| J, Dayani NE, Franke L, Fuyuno Y, Hart A, Juyal RC, Juyal G | | | | | |
|---|---|---|---|---|---|
| Cortes A, Hadler J, Pointon JP, Robinson PC, Karaderi T, Leo P, Cremin K, Pryce K, Harris J, Lee S, Joo KB, Shim SC, Weisman M, Ward M, Zhou X, Garchon HJ, Chiocchia G, Nossent J, Lie BA, Førre Ø, Tuomilehto J, Laiho K, Jiang L, Liu Y, Wu X, Bradbury LA, Elewaut D, Burgos-Vargas R, Stebbings S, Appleton L, Farrah C, Lau J, Kenna TJ, Haroon N, Ferreira MA, Yang J, Mulero J, Fernandez-Sueiro JL, Gonzalez-Gay MA, Lopez-Larrea C, Deloukas P, Donnelly P | 2013 | Identification of multiple risk variants for ankylosing spondylitis through high-density genotyping of immune-related loci" | https://www.ebi.ac.uk/gwas/publications/23749187 | NHGRI-EBI GWAS Catalog, 23749187 |
| Hinks A, Cobb J, Marion MC, Prahalad S, Sudman M, Bowes J, Martin P, Comeau ME, Sajuthi S, Andrews R, Brown M, Chen WM, Concannon P, Deloukas P, Edkins S, Eyre S, Gaffney PM, Guthery SL, Guthridge JM, Hunt SE, James JA, Keddache M, Moser KL, Nigrovic PA, Onengut-Gumuscu S, Onslow ML, Rosé CD, Rich SS, Steel KJ, Wakeland EK, Wallace CA, Wedderburn LR, Woo P, Bohnsack JF, Haas JP, Glass DN, Langefeld CD, Thomson W, Thompson SD | 2013 | Dense genotyping of immune-related disease regions identifies 14 new susceptibility loci for juvenile idiopathic arthritis | https://www.ebi.ac.uk/gwas/publications/23603761 | NHGRI-EBI GWAS Catalog, 23603761 |
| Okada Y, Wu D, Trynka G, Raj T, Terao C, Ikari K, Kochi Y, Ohmura K, Suzuki A, Yoshida S, Graham RR, Manoharan A, Ortmann W, Bhangale T, Denny JC, Carroll RJ, Eyler AE, | 2013 | Genetics of rheumatoid arthritis contributes to biology and drug discovery | https://www.ebi.ac.uk/gwas/publications/24390342 | NHGRI-EBI GWAS Catalog, 24390342 |

| | | | | |
|---|---|---|---|---|
| Greenberg JD, Kremer JM, Pappas DA, Jiang L, Yin J, Ye L, Su DF, Yang J, Xie G, Keystone E, Westra HJ, Esko T, Metspalu A, Zhou X, Gupta N, Mirel D, Stahl EA, Diogo D, Cui J, Liao K, Guo MH, Myouzen K, Kawaguchi T, Coenen MJ | | | | |
| Beecham AH, Patsopoulos NA, Xifara DK, Davis MF, Kemppinen A, Cotsapas C, Shah TS, Spencer C, Booth D, Goris A, Oturai A, Saarela J, Fontaine B, Hemmer B, Martin C, Zipp F, D'Alfonso S, Martinelli-Boneschi F, Taylor B, Harbo HF, Kockum I, Hillert J, Olsson T, Ban M, Oksenberg JR, Hintzen R, Barcellos LF, Agliardi C, Alfredsson L, Alizadeh M, Anderson C, Andrews R, Søndergaard HB, Baker A, Band G, Baranzini SE, Barizzone N, Barrett J, Bellenguez C, Bergamaschi L, Bernardinelli L, Berthele A, Biberacher V, Binder TM, Blackburn H, Bomfim IL, Brambilla P, Broadley S, Brochet B, Brundin L, Buck D, Butzkueven H, Caillier SJ, Camu W, Carpentier W, Cavalla P, Celius EG, Coman I, Comi G, Corrado L, Cosemans L, Cournu-Rebeix I, Cree BA, Cusi D, Damotte V, Defer G | 2013 | Analysis of immune-related loci identifies 48 new susceptibility variants for multiple sclerosis | https://www.ebi.ac.uk/gwas/publications/24076602 | NHGRI-EBI GWAS Catalog, 24076602 |
| Beecham AH, Patsopoulos NA | 2019 | Downloadable genotypes of present-day and ancient DNA data (compiled from published papers) V37.2 | https://reich.hms.harvard.edu/allen-ancient-dna-resource-aadr-downloadable-genotypes-present-day-and-ancient-dna-data | David Reich Lab published data amh_repo V37.2, V37.2 |
| Bakker OB | 2018 | GWAS round 2 | http://www.nealelab.is/uk-biobank | UK Biobank, uk-biobank |

Bakker OB        2019   500FG Cytokine QTL data          https://molgenis26.gcc.        The 1000ibd
                                                       rug.nl/downloads/            MOLGENIS database,
                                                       500FG/cytokines.zip          500FG/cytokines

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
