## [Decision Letter]

**Acceptance summary:**

Dominguez-Andrés et al., collect a large amount of immune-related trait association data from a cohort is made up of 534 individuals of Western European ancestry. The goal was to track the evolutionary trajectories of cytokine production capacity over time in a number of patients with different exposure to infectious organisms, infectious disease, autoimmune and inflammatory diseases using the 500 Functional Genomics cohort of the Human Functional Genomics Project. From this analysis it was hypothesized that the Neolithic transition was characterized by strong changes in the adaptive response to pathogens in human biology.

**Decision letter after peer review:**

Thank you for submitting your article "Evolution of cytokine production capacity in ancient and modern European populations" for consideration by *eLife*. Your article has been sent to 3 peer reviewers, one of whom is a member of our Board of Reviewing Editors and the evaluation has been overseen by Mone Zaidi as the Senior Editor. The following individual involved in review of your submission has agreed to reveal their identity: Fernando Racimo (Reviewer #3).

The Reviewing Editor has drafted this to help you prepare a revised submission.

Essential Revisions:

Overall, the main concern of both reviewers focused on statistical methodology followed by the authors and its limitations. There was a feeling that it was not strong enough to support the manuscript's conclusions as stated. These reviewer comments, which were numerous, centered on what was felt to be inadequate models and partitioning. Fully addressing the reviewer's concerns and revising the model and data analysis as appropriate will be needed to strengthen the paper.

1. The authors should comment on the impact and limitations on using the "average genotyped dosage" to be substituted for missing variants and data interpretation. It was described that the dosage of these variants was multiplied by their effect size while the dosage of missing variants in a sample were supplemented with the average dosage. Although agreed that this method doesn't skew the PRS preferentially one way or the other, but it could dilute or strengthen the scale of the data without knowing what the "true" values of the missing data points are.

2. Peripheral blood mononuclear cells were challenged with bacterial, fungal, viral, and non-microbial stimuli while measuring 6 cytokines (TNF α, IL-1 β, IL-6, IL-17, IL-22, and IFN γ) at 24 hrs and 7 days post-stimulation yielding 105 unique cytokine-stimulation pairs. Although a large number of such pairings were provided, likely due to data availability, the data set is not complete i.e. for the same organism not all 6 cytokines were followed and not all pairings had both a 24 hour and a 7 day time points. Some had a 48 hour time point. The authors should try to provide a subset of data in which there is completeness of data to allow the readers to more clearly be able to compare trends appropriately (i.e. across the same cytokine markers or the same time point post stimulus etc …).

3. In Figure 1, how was this particular subset of cytokines-stimulants pairings chosen from the larger set in Figure S1? The authors should show representative examples that highlights their discussion in to the text and to clearly label as such intracellular vs. extracellular organisms, zoonotic organisms vs not, etc. For example, the text makes reference to data on the IFN γ responses which are not presented in Figure 1.

4. The authors should also comment on the impact of inter-patient or inter-measurement variabilities of the cytokine responses to stimuli could impact their current results. They should describe any potential limitations or biases that this might induce in their data set.

5. The authors estimated cytokine production capacity based on PRS which authors claim to have significant differences between populations in various historical periods with the largest strength difference reported before and after the Neolithic revolution. In Figure 1, the split was done to look at before and after the Neolithic era and the linear regression correspond to those two eras. However, the authors do not comment or show the data to demonstrate why they choose that specific break point as opposed to looking at every historical era transition i.e. from early upper paleolithic to late upper paleolithic to Mesolithic to Neolithic to post-Neolithic to modern. The authors should justify why grouping together all of the historical eras to being before and after the Neolithic revolution is acceptable and demonstrate from the data that the linear regression lines would follow the same pattern in each distinct historical period vs in the pre- and post-Neolithic grouping. This should be addressed in the main figures or as a supplemental figure.

6. In Figure 1-3 and as briefly discussed in the text, there are very few data points in the pre-Neolithic in comparison to post-era which would make it challenging for the reader to draw robust conclusions about the trends in the pre-Neolithic era. This should be addressed in the main text to prevent any overinterpretation of the main figures. In the supplemental figure S2, the authors do make claims about robustness of the trend when comparing to downsampled numbers in the post-Neolithic era and do comment that the power decreases with lower number of samples.

7. Throughout Figures 1-4, the authors state that the correlation pattern is the same across a number of thresholds 0.96, 0.9, 0.8, and 0.7. It would be best for the only one threshold range to be presented in these main figures and to move the complete data across multiple threshold ranges into a supplemental figure as the added ranges do not add anything new to the data results and interpretation.

8. Although the authors picked a very homogeneous population, the authors should discuss in terms of addressing the limitations of the current study, what the impact of other factors or events throughout history that could have a role here such as the development of vaccines the cross reactivity of antigens that may be shared between organisms, the development of medications (antimicrobials, antivirals, anti-inflammatory, etc …)?

9. Although the authors had a robust discussion on the prevalence of inflammatory and autoimmune diseases within the European population over different historical eras. A similar discussion around factors such as population density, migration, living conditions, living environments, diet/nutrition, sanitation/hygiene, etc … is lacking in discussing the evolution of cytokine production to specific pathogens.

10. The authors should highlight additional limitations of this current study in terms of the generalizability to other populations or to clearly state that this is limited to the European population at the specified latitude and longitudes used. The authors did demonstrate for the most part agreement with prior published studies which adds strength to their current results.

11. A concern is that the statistical methodology followed by the authors is not appropriate to support their conclusions – in particular, the fact that data imbalance and their effect on the P-values has not been properly accounted for, and that the choice of pre-partitioned pair of linear models is not the correct way to detect a shift in scores (or to make the much stronger claim of adaptation). I would recommend changing the model being used so as not to rely on a preparation of the data, and to properly account for genetic drift.

---

## [Author Response]

1. The authors should comment on the impact and limitations on using the "average genotyped dosage" to be substituted for missing variants and data interpretation. It was described that the dosage of these variants was multiplied by their effect size while the dosage of missing variants in a sample were supplemented with the average dosage. Although agreed that this method doesn't skew the PRS preferentially one way or the other, but it could dilute or strengthen the scale of the data without knowing what the "true" values of the missing data points are.

We thank the reviewer for this important remark. Replacing missing values by the average dosage would indeed dilute the observed effects or artificially increase the power of the observed trends. The alternative would be using only SNPs present in all samples. This would also bias our analysis since older samples have a higher error rate during genotype calling compared to our more modern samples. This is due to the sparsity of data and cytosine deamination which affects older samples to a greater extent. In our analysis, we tried to alleviate potential bias by using multiple thresholds for average missing genotype rate. By focusing on the consistent pattern across different thresholds, we were able to put our findings in a better context and show that the trends we observed hold true even when filtering out samples with high rates of substitute values. We have included text in the Discussion part of revised manuscript (lines 115 to 119).

2. Peripheral blood mononuclear cells were challenged with bacterial, fungal, viral, and non-microbial stimuli while measuring 6 cytokines (TNF α, IL-1 β, IL-6, IL-17, IL-22, and IFN γ) at 24 hrs and 7 days post-stimulation yielding 105 unique cytokine-stimulation pairs. Although a large number of such pairings were provided, likely due to data availability, the data set is not complete i.e. for the same organism not all 6 cytokines were followed and not all pairings had both a 24 hour and a 7 day time points. Some had a 48 hour time point. The authors should try to provide a subset of data in which there is completeness of data to allow the readers to more clearly be able to compare trends appropriately (i.e. across the same cytokine markers or the same time point post stimulus etc …).

We thank the reviewer for these remarks. The stimulation periods were chosen based on extensive studies that showed that the timepoints used were best suited for assessing monocyte-derived and lymphocyte-derived cytokines per stimulus. Not all the stimuli induce the production of all cytokines, so the selection of the cytokine-stimulus pairs was performed for those pairs in which a cytokine production could be measured (PMID: 1385767; PMID: 19380112; PMID: 27814509; PMID: 27814508; PMID: 27814507). The missing stimulus-cytokine pairs are simply due to the fact that some stimuli did not induce the production of some cytokines. The differences in the cytokine availability and time points are adjusted to the optimal time of production per stimuli. Monocyte-derived cytokines (IL-1b, IL-6 and TNFa) are early response cytokines, produced by innate immune cells shortly after stimulation (6 to 24h, the latter being used standard). IFNγ, IL-17 and IL-22 are lymphocyte-derived cytokines, produced by adaptive immune cells, in this case T helper cells upon stimulation by antigen-presenting cells. These cells need to differentiate for 2-7 days before they start to produce these cytokines, this is the reason why the time point of the measurements of these cytokines is 7 days. In the case of IFNγ, the kinetics of its production is more rapid, and we therefore measured it 48h after stimulation in whole blood samples.

We have included these considerations and new references in the new version of the text (lines 82 to 87).

3. In Figure 1, how was this particular subset of cytokines-stimulants pairings chosen from the larger set in Figure S1? The authors should show representative examples that highlights their discussion in to the text and to clearly label as such intracellular vs. extracellular organisms, zoonotic organisms vs not, etc. For example, the text makes reference to data on the IFN γ responses which are not presented in Figure 1.

The selected cQTL traits were chosen based on whether their effects were consistently significant and in the same direction across different threshold combinations. In the new version of the text and the figures we have highlighted in the figure legends which of the organisms are intracellular/extracellular and which organisms cause zoonotic diseases (lines 419-421).

4. The authors should also comment on the impact of inter-patient or inter-measurement variabilities of the cytokine responses to stimuli could impact their current results. They should describe any potential limitations or biases that this might induce in their data set.

The interindividual variability in the responses is something that happens naturally and is part of the analysis and the representations performed. Therefore, the variability of the cytokine responses between different individuals caused by the intrinsic gene expression variation and the environmental factors cannot be avoided. In this study we have used a cohort of individuals with a similar genetic background (Western-European ancestry) but with balanced characteristics in terms of age, sex, BMI and habits (PMID: 27814508), which allowed us to represent the natural variability in the responses. Cytokines were measured in batches in which a certain cytokine was measured the same day for all the samples. This was done to decrease the potential technical variations. Moreover, in a recent study we evaluated the intra-individual variation of cytokine production capacity, which was found limited (ter Horst et al., J Immunol, in press). On the other hand, inter-individual variability of cytokine production was largely depedent on genetic variants (PMID: 27814507, PMID: 29784908) We have included these considerations in the new version of the manuscript (lines 96 to 105).

5. The authors estimated cytokine production capacity based on PRS which authors claim to have significant differences between populations in various historical periods with the largest strength difference reported before and after the Neolithic revolution. In Figure 1, the split was done to look at before and after the Neolithic era and the linear regression correspond to those two eras. However, the authors do not comment or show the data to demonstrate why they choose that specific break point as opposed to looking at every historical era transition i.e. from early upper paleolithic to late upper paleolithic to Mesolithic to Neolithic to post-Neolithic to modern. The authors should justify why grouping together all of the historical eras to being before and after the Neolithic revolution is acceptable and demonstrate from the data that the linear regression lines would follow the same pattern in each distinct historical period vs in the pre- and post-Neolithic grouping. This should be addressed in the main figures or as a supplemental figure.

We thank the reviewer for this helpful suggestion. We have followed it and made the following changes:

– The original figures showed only models using two separate linear regression lines and the different thresholds for missing genotype rates showed consistent results.

– In the new figures we depict LOESS regression models to better show the difference in mean PRS at every point in time and we additionally show boxplots with the different major age periods pooling the paleolithic and mesolithic samples together as pre-neolithic samples in order to account for the lower sample number in the earlier historical periods. To highlight this we have added a new section in lines 123 to 129.

– In the revised figure 2 we add LOESS regression models for which we do not bias our analysis into defining a break at a certain time period. We furthermore show boxplots with pairwise comparisons (student’s T-test) for broader time periods highlighting the changes in PRS that would correspond with major changes in human lifestyle such as the shift from a hunter-gatherer to a neolithic lifestyle or the later urbanization of modern human societies.

– In the new version of Figure 3 we confirm that the various traits showing a clear change in PRS start at the advent of the Neolithic or post-Neolithic era using both the LOESS regression and pairwise comparisons (student T-test).

– Similarly the heatmap in our original figure 4 has also been revised to only show the large sample set.

6. In Figure 1-3 and as briefly discussed in the text, there are very few data points in the pre-Neolithic in comparison to post-era which would make it challenging for the reader to draw robust conclusions about the trends in the pre-Neolithic era. This should be addressed in the main text to prevent any overinterpretation of the main figures. In the supplemental figure S2, the authors do make claims about robustness of the trend when comparing to downsampled numbers in the post-Neolithic era and do comment that the power decreases with lower number of samples.

We fully agree with the reviewer that the power decreases with lower number of samples for the pre-Neolithic period. We have carefully checked whether our conclusion was robust by using a down-sampling method. Previously we plotted the correlation coefficients of the full dataset compared to the downsampled dataset. In the new Figure 4—figure supplement 1 we plot the -log10 P values times the sign of the correlation coefficient of the traits in the main figures using both the full dataset as well as the downsampled dataset to better show the consistency in direction of the effects. We have also discussed this more in depth on lines 267-272.

7. Throughout Figures 1-4, the authors state that the correlation pattern is the same across a number of thresholds 0.96, 0.9, 0.8, and 0.7. It would be best for the only one threshold range to be presented in these main figures and to move the complete data across multiple threshold ranges into a supplemental figure as the added ranges do not add anything new to the data results and interpretation.

We followed this suggestion and revised the figures accordingly. Please see details described in our response to point number 5.

8. Although the authors picked a very homogeneous population, the authors should discuss in terms of addressing the limitations of the current study, what the impact of other factors or events throughout history that could have a role here such as the development of vaccines the cross reactivity of antigens that may be shared between organisms, the development of medications (antimicrobials, antivirals, anti-inflammatory, etc …)?

We thank the reviewer for raising such an important point. We have included a Discussion about these topics in the new version of the manuscript (lines 273 to 282). We have to mention though that the analyses in the 500FG were performed in individuals who did not use any medication.

9. Although the authors had a robust discussion on the prevalence of inflammatory and autoimmune diseases within the European population over different historical eras. A similar discussion around factors such as population density, migration, living conditions, living environments, diet/nutrition, sanitation/hygiene, etc … is lacking in discussing the evolution of cytokine production to specific pathogens.

We agree that this is a very good point raised by the reviewer, and we added that in the revised manuscript. As mentioned also in our response to the previous reviewer, in this study we focus on the effects of different pathogens in cytokine production and therefore in the response to different infections. Human evolution has been influenced by multiple factors, such as migrations, urbanization, climate, diet, which also influence the responses to pathogens. The advent of the Neolithic era was a milestone for human civilization and human lifestyle, with higher density of human populations and rate of contact with animals which increased the emergence of pathogens. Our data show that despite the fact that other events and conditions throughout human history have influenced our immune responses to pathogens, the advent of the Neolithic was a turning point for our capacity to respond to pathogens. The evolution of cytokine production to specific pathogens is also a very important factor to take into account. The more contact humans have had with a pathogen, the greater the likelihood that the interaction between host and pathogen will lead to changes in the cytokine production and also the adaptation of the pathogen to the human host. We have included a Discussion about these topics in the new version of the manuscript (lines 273 to 282).

10. The authors should highlight additional limitations of this current study in terms of the generalizability to other populations or to clearly state that this is limited to the European population at the specified latitude and longitudes used. The authors did demonstrate for the most part agreement with prior published studies which adds strength to their current results.

Indeed, this is a very valid point. In our study we focus on summary statistics obtained from European populations and only employ European aDNA samples, so our results should not be extrapolated to other populations from other geographical areas. We have included this in the Discussion of the new version of the manuscript (lines 289 to 292). However, as the reviewer mentions, our findings are mostly in agreement with previous studies in other populations, which adds robustness to the results of our study.

11. A concern is that the statistical methodology followed by the authors is not appropriate to support their conclusions – in particular, the fact that data imbalance and their effect on the P-values has not been properly accounted for, and that the choice of pre-partitioned pair of linear models is not the correct way to detect a shift in scores (or to make the much stronger claim of adaptation). I would recommend changing the model being used so as not to rely on a preparation of the data, and to properly account for genetic drift.

We thank the reviewer for this important remark. In order to account for the data imbalance in terms of samples per historical age period we pooled all samples prior to the neolithic age together before testing the difference in average PRS between neighbouring periods using a T test, we also show LOESS regression models to not depend on a pre-partitioning of our data. In addition we aim to show how consistent our post-Neolithic results are by downsampling the post-Neolithic samples and reperforming linear regression (please see our replies to points 5 and 6).

In addition we performed a seperate test for adaption as opposed to genetic drift using Wright’s fixation index (Fst) to test for significant differences in genetic differentiation of trait specific SNPs compared to background SNPs in order to determine whether the changes we see in PRS over time are due to selection or due to genetic drift. This analysis shows that changes between the pre-Neolithic to Neolithic are not significantly different from drift, whereas after the onset of the Neolithic we do see significant amount of selection. We have explained this further in the manuscript line 130 to 135 and included the new Table S2.